# A Rotary Spacer System for Energy-Efficient Membrane Fouling Control in Oil/Water Emulsion Filtration

**DOI:** 10.3390/membranes12060554

**Published:** 2022-05-26

**Authors:** Normi Izati Mat Nawi, Afiq Mohd Lazis, Aulia Rahma, Muthia Elma, Muhammad Roil Bilad, Nik Abdul Hadi Md Nordin, Mohd Dzul Hakim Wirzal, Norazanita Shamsuddin, Hazwani Suhaimi, Norhaniza Yusof

**Affiliations:** 1Department of Chemical Engineering, Universiti Teknologi PETRONAS, Seri Iskandar 32610, Perak, Malaysia; normi_16000457@utp.edu.my (N.I.M.N.); afiq.laziz@hotmail.com (A.M.L.); nahadi.sapiaa@utp.edu.my (N.A.H.M.N.); mdzulhakim.wirzal@utp.edu.my (M.D.H.W.); 2Chemical Engineering Department, Lambung Mangkurat University, Banjarbaru 70714, South Kalimantan, Indonesia; arahma@mhs.ulm.ac.id (A.R.); melma@ulm.ac.id (M.E.); 3Doctoral Program of Environmental Science, Postgraduate Program, Lambung Mangkurat University, Jl Brigjen H. Hasan Basri, Kayutangi, Banjarmasin 70123, South Kalimantan, Indonesia; 4Faculty of Integrated Technologies, Universiti Brunei Darussalam, Gedung BE1410, Brunei; norazanita.shamsudin@ubd.edu.bn (N.S.); hazwani.suhaimi@ubd.edu.bn (H.S.); 5Advanced Membrane Technology Research Centre (AMTEC), School of Chemical and Energy Engineering, Faculty of Engineering, Universiti Teknologi Malaysia, UTM Johor Bahru, Skudai 81310, Johor, Malaysia; norhaniza@petroleum.utm.my

**Keywords:** membrane fouling, dynamic membrane filtration, rotating spacer, oil/water emulsion

## Abstract

Membrane fouling deteriorates membrane filtration performances. Hence, mitigating membrane fouling is the key factor in sustaining the membrane process, particularly when treating fouling-prone feed, such as oil/water emulsions. The use of spacers has been expanded in the membrane module system, including for membrane fouling control. This study proposed a rotating spacer system to ameliorate membrane fouling issues when treating an oil/water emulsion. The system’s effectiveness was assessed by investigating the effect of rotating speed and membrane-to-disk gap on the hydraulic performance and the energy input and through computational fluid dynamics (CFD) simulation. The results showed that the newly developed rotary spacer system was effective and energy-efficient for fouling control. The CFD simulation results proved that the spacer rotations induced secondary flow near the membrane surface and imposed shear rate and lift force to exert fouling control. Increasing the rotation speed to an average linear velocity of 0.44 m/s increased the permeability from 126.8 ± 2.1 to 175.5 ± 2.7 Lm^−2^h^−1^bar^−1^. The system showed better performance at a lower spacer-to-membrane gap, in which increasing the gap from 0.5 to 2.0 cm lowered the permeability from 175.5 ± 2.7 to 126.7 ± 2.0 Lm^−2^h^−1^bar^−1^. Interestingly, the rotary system showed a low energy input of 1.08 to 4.08 × 10^−3^ kWhm^−3^ permeate when run at linear velocities of 0.27 to 0.44 ms^−1^. Overall, the findings suggest the competitiveness of the rotary spacer system as a method for membrane fouling control.

## 1. Introduction

Membrane-based processes are leading water and wastewater treatment technologies due to their high selectivity, low energy consumption, minimal chemical usage, and ease of scaling up [1,2,3]. However, their widespread application is hindered by membrane fouling, which may escalate the cost of system maintenance and operation. Membrane fouling occurs due to the accumulation of foulant materials on the membrane surface that fully/partially block the pores, thereby decreasing water permeation over time. The deposited foulant, such as suspended particles, colloids, and microorganisms, may build up a cake layer on the membrane surface, exacerbating the membrane performance and causing the rapid decline of permeability over time [4]. Therefore, proper control of membrane fouling is required to ensure process sustainability. The most common attempts include membrane surface modification [5,6], optimization of operational parameters [7], and pretreatment of feed [8,9].

Dynamic mechanical filtration, known as shear-enhanced filtration, has been utilized as one of the fouling control strategies. It can be achieved in many ways, such as by rotating [10,11,12], vibrating [13,14,15], or reciprocating the motion of the membrane modules [16,17]. In a conventional cross-flow filtration system, high shear rates at the feed side of the membrane surface can be achieved by increasing the tangential fluid velocity along the membrane. However, it is energy-intensive to pump and overcome the loss of feed pressure along the membrane [18]. Thus, dynamic filtration has become a promising alternative to the cross-flow system since it can substantially improve the throughput by reducing the fouling effect. Despite the advantages, it was found that the widespread use of these systems is challenged by their complexity, and they may require additional construction and maintenance and have a limited capacity for membrane area. For instance, Spintek (Los Alamitos, CA, USA) only allows 2.3 m^2^ of membrane area in their module, and there is a limit of 5 m^2^ for CRD from Novoflow (Rain, Germany) and 8 m^2^ for Dyno from Bokela GmbH (Karlsruhe, Germany) [19]. Hence, these systems may not be competitive when large membrane areas are required, but are indeed quite favourable for niche applications.

The hydrodynamic effect in the filtration system is expected to increase permeate flux by reducing concentration polarisation and cake build-up [20]. The reduction of concentration polarisation also extends the pressure range, favouring higher fluxex. It is worth noting that the mechanical movement generates a secondary flow of liquid that disturbs the mass transfer boundary layer and promotes local mixing near the membrane surface [10]. For instance, Jaffrin et al. [21] showed that incorporating a rotating disk equipped with vanes in the filtration system achieved the highest permeate fluxes thanks to its high shear rates. Another recent study that applied the dynamic filtration of vibratory shear-enhanced processing (VSEP) concept was reported by Hapońska et al. [22] for microalgae dewatering. It was found that the membranes yielded a higher permeate volume and reduced the irreversible membrane fouling in the dynamic filtration system compared to the conventional cross-flow filtration. At the same time, the dynamic filtration was economically beneficial since the improved flux or productivity reduced the membrane area [23].

Frappart et al. [24] studied the hydrodynamic effects on the ultrafiltration of microalgae suspension by comparing the dynamic filtration module with cross-flow filtration. The results revealed that the module equipped with a rotating disk yielded a permeate flux that was almost two times higher than that of the cross-flow system. Sarkar et al. [25] investigated the effect of operating parameters of casein whey ultrafiltration in a rotating disk filtration module compared to the stationary membrane. It was observed that the dynamic rotating disk filtration achieved 38.7% higher flux at 300 rpm compared to the stationary system. The findings suggest the importance of shear across the membrane in minimizing the effects of concentration polarisation. The efficiency of dynamic filtration has also been proven through simulation studies scrutinizing the system’s feasibility [26,27]. Membrane module rotation could also achieve a dynamic system [28]. In recent reports, a spacer is seen as more practical in creating the flow of liquid near the membrane surface [29]. The spacer can be made from low-density material. It can be designed thin to maintain a high membrane module packing density [14] and designed as an inherent part of the membrane module system [30].

Moreover, this configuration can achieve a high rejection rate and improved membrane selectivity. Choi et al. [31] observed that the removal of 2-methylisoborneol and Geosmin from drinking water in the Han River (Korea) by nanofiltration membranes was enhanced by the introduction of flow shear in the membrane systems. 2-methylisoborneol and Geosmin are mainly responsible for global taste and odour changes in drinking water. An increase in shear rates near the membrane surface resulted in an improvement in membrane selectivity. Wu et al. [29] also suggested that spacers can be synthesized from adsorbents, for example, to improve the removal of micropollutants from wastewater. Biological adsorbents can be incorporated with spacers to support biofilm growth for enhanced removal efficiencies in dynamic membrane modules.

This study proposed a submerged membrane filtration system equipped with a rotating spacer to confront membrane fouling issues while treating an oil/water emulsion using laboratory-made polyvinylidene fluoride (PVDF). The developed system was designed to delay the fouling effect and thus reduce the system’s energy consumption. The rotating motion of the spacer created continuous momentum that reduced the energy input, as later demonstrated by the present study results. Several studies on operating conditions such as rotating speed and membrane-to-disk gap were thoroughly performed to investigate the effect of these operating conditions on the membrane hydraulic performance. Finally, computational fluid dynamics (CFD) was conducted to visualize the wall shear stress distribution on the rotating spacer and the static membrane.

## 2. Materials and Methods

### 2.1. Membrane Preparation and Characterization

In order to assess the fouling control exerted by the rotary spacer, a series of filtration tests were conducted using laboratory-made polyvinylidene fluoride (PVDF) membrane. The membrane was fabricated via the non-solvent-induced phase separation from a dope solution composed of 15 wt.% PVDF (Sigma Aldrich, St. Louis, MO, USA) as the polymer, 1 wt.% polyethylene glycol (PEG, Sigma Aldrich, USA) as the additive, and 84 wt.% dimethylacetamide (DMAC, Sigma Aldrich, USA) as the solvent. The polymer, additive, and solvent were mixed using a magnetic stirrer for 24 h, or until the solution became homogeneous. The dope solution was then left idle for 24 h to ensure the release of all entrapped air bubbles. Subsequently, it was cast with a wet thickness of 210 µm on nonwoven support (Novatexx 24414, Freudenberg Filtration Technologies, Germany) as backing material to provide mecha nical support to the thin polymer film [32]. The cast film was immediately immersed in a water bath, acting as the non-solvent to allow the phase inversion. The formed membrane was left in the bath overnight to ensure all the trace solvent was removed entirely. The membrane was stored wet in a water container at room temperature until usage.

The properties of the PVDF membrane were determined as follows. The morphology, pore size distribution, thickness, and surface contact angle were determined using a scanning electron microscope (SEM, ZEISS, Oberkochen, Germany), capillary flow porometer (CFP, Porolux, Belgium), electronic digital micrometre screw gauge (Mitutoyo, Kanagawa, Japan), and goniometer (DataPhysics, Fildestadt, Germany), respectively.

### 2.2. Oil-in-Water Emulsion Feed Preparation

The oil-in-water (oil/water) emulsion was used as feed to evaluate the magnitude of the membrane fouling control provided by the rotary spacer system. The feed sample was prepared by mixing a crude oil (obtained from one of the crude oil wells in Malaysia) in deionized water with a small amount of synthesis-grade sodium dodecyl sulfate (SDS 98, Sigma Aldrich, St. Louis, MO, USA) with a ratio of 1:9 (wt./wt.) of SDS to the crude oil. The mixture with a total volume of 10 L was stirred for two days until a stable emulsion was obtained. The added SDS acted as a surfactant to stabilize the oil-in-water emulsion. The oil concentration in the feed was fixed at 1000 ppm. A small volume of feed sample (10 mL) was subsequently analyzed using particle size and zeta potential analyzer (Malvern, Zetasizer Nano ZSP, Malvern, UK) to map the distribution of the oil droplets. The droplet size was analysed using the dynamic light scattering method by assuming all the detected particles were the oil-in-water droplets in spherical shapes.

### 2.3. Membrane Panel Assembly and Filtration Set-Up

Figure 1 illustrates the laboratory-scale submerged filtration set-up equipped with a rotary disk spacer placed between two adjacent membrane panels. The spacer was in a circular disk with a thickness of 2 mm attached at the centre point to a rotating shaft. The shaft rotation was driven by a motor equipped with a rotation speed regulator. The membrane panel had an effective surface area of 0.2 m^2^ in the shape of a segment of a half-circle with a diameter of 17 cm, which was connected to a permeate collecting system. A vacuum pump was used to drive permeation at a constant pressure of −0.1 bar. The filtration tests were conducted for 120 min and comprised 12 cycles. Each cycle consisted of nine minutes of permeation and one minute of relaxation. During the relaxation, no vacuum was exerted and the system was left idle temporarily. During the relaxation, the volume of the permeate collected during the permeation was measured then returned to the feed tank to maintain the feed condition (i.e., feed level and composition). About 10 mL of the permeate was stored for oil rejection analysis.

### 2.4. Filtration Test

As a proof-of-concept report on the effectiveness of a rotary spacer in controlling membrane fouling, two parameters were assessed: the effect of rotating speed and membrane-to-disk gap. Each filtration was conducted in duplicate, and the results are presented as average ± standard deviation. The permeate volume was used to calculate filtration flux (*J*, Lm^−2^h^−1^) and membrane permeability (*L*, L m^−2^ h^−1^ bar^−1^) using Equation (1) and Equation (2), respectively.
(1)$$J=\frac{V}{At}$$
(2)$$L=\frac{J}{\Delta P}$$
where *V* is permeated volume (L), *A* is membrane total effective filtration area (m^2^), *t* is filtration time (h), and ∆*P* is trans-membrane pressure (bar). The filtration time was calculated as the sum of the filtration and the relaxation. As such, the presented permeability data are the net value. The system was evaluated by using various rotation speeds of the spacer, 0, 30, 35, 40, 45, and 50 rpm, and membrane-to-disk gaps of 0.5, 1.0, 1.5, and 2.0 cm.

### 2.5. Estimation of Energy Consumption

In order to evaluate the energy consumption of the developed rotary spacer, the energy input for mechanical rotation of a hypothetically full-scale set-up was projected. The full-scale module was assumed to be a circular panel with a diameter of 2.0 m and thickness of 4 × 10^−3^ m, with the membrane sheets mounted on both sides. The membrane panel was set static, sandwiched by rotating disks with equal diameters. The rotation speeds of the hypothetical full-scale module were set at the same values as the average linear velocities used in the experiments. The rotating shaft power (*P_R_*, kW) and specific filtration energy for the rotary spacer system (*S_ERT_*, kWhm^−3^) were estimated using Equations (3)–(6).
(3)$$P_{R}=\frac{1}{2}\;C_{D}\;\rho_{F}\;A_{P}\;v_{R}^{3}\times 10^{-3}$$
(4)$$A_{P}=d_{D}t_{D}$$
(5)$$v_{R}=\frac{d_{E}}{d_{D}}\;r_{D}\;\omega_{E}$$
(6)$$S_{ERT}=\eta \frac{P_{R}}{{\dot{V}}_{P}}$$
where $C_{D}$ is the drag coefficient for the thin cylinder (1.15), $A_{P}$ is the projected area (m^2^), $v_{R}$ is the average linear velocity of the rotating disk (ms^−1^), $d_{D}$ is the diameter of the full-scale rotating disk (2 m), $t_{D}$ is the thickness of the full-scale rotating disk (4 × 10^−3^ m), $d_{E}$ is the diameter of the experimental rotating disk (0.17 m), $r_{D}$ is the the disk diameter (m), $\omega_{E}$ is the angular velocity of the experimental disks (rads^−1^), and ${\dot{V}}_{P}$ is the volumetric flow rate of the permeate (m^3^h^−1^).

### 2.6. Data Analysis

The results were analyzed statistically. The significance of the effect of each operating condition on overall permeability in each parameter was evaluated using a one-way analysis of variance at 95% confidence intervals (*p*-value < 0.05). Then, Tukey’s honest significant difference post hoc analyses were performed to identify multiple pairs of mean values when evaluating more than two data points [33].

### 2.7. CFD Simulation

The fluid dynamic was visualized using using Ansys fluent (2020 R2, Student Version) by employing the Navier–Stokes and the continuity equations (Equations (7) and (8)) coupled using the SIMPLE algorithm. The domains were discretized into rectangular elements and solved using the finite element method and by assuming the no-slip condition on the walls.
(7)∇·u=0
(8)$$\rho \;\left(u\cdot \nabla u\right)=-\nabla p+µ\nabla^{2}u$$
where u is the velocity vector, p is the pressure, ρ is the water density (kgm^−3^), and µ is the water dynamic viscosity (kgm^−1^s^−1^).

The optimum mesh size was determined using the sensitivity analysis. The meshing sizes were varied from 1 to 5 (Table 1), and their qualities as well as the velocity plots are shown in Figure 2. Eventually, mesh 4 was taken as the optimum and was used for the rest of the simulation throughout the study.

## 3. Results and Discussion

### 3.1. Membrane and Feed Properties

The membrane sheet, together with the baking support, had an overall thickness of 261 ± 6.1 μm, mean pore size of 0.11 μm, top surface contact angle of 67.9 ± 1.0°, and clean water permeability of 461.9 ± 26.5 Lm^−2^h^−1^bar^−1^. Figure 3 shows the pore size distribution and the SEM images of the developed membrane. The properties met the requirement of membranes for oil/water separation [34]. It was hydrophilic, and the nominal pore sizes of the PVDF membrane were much smaller than the oil droplet sizes. It must be noted that the present study focuses on assessing the performance of the rotary spacer on membrane fouling control. Hence, a detailed analysis of the membrane characteristics was beyond the scope.

The sizes of the oil droplets in the oil/water emulsion system were in a three-modal distribution with the peaks at 0.25, 0.9, and 4.0 µm (Figure 4). The synthesized oil/water emulsion was found stable during the experiment duration for about 2 months without any floating layer. Judging from the oil droplet size distribution and by comparing with the membrane’s pore size in Figure 3, all clusters of droplets should be retained by the membrane because they were much larger than the mean pore size of the membrane (0.11 µm). The membrane was expected to fully retain the oil droplets from the obtained data on the membrane properties.

### 3.2. Effect of Rotating Speed

The primary role of the rotary spacer in the proposed system was that the rotation induced a secondary flow of liquid that later exerted shear on the membrane surface and eventually increased the flux. Figure 5 summarizes the effect of the rotating speed of the disk spacers on the oil/water emulsion permeability at a constant membrane-to-disk gap of 1.5 cm. Figure 5a shows the effect of rotation speed on the membrane permeability with respect to filtration time. The unavoidable membrane fouling resulted in a 45–55% permeability loss from the initial value, observed for all tests before reaching their quasi-steady-state value (as summarized in Figure 5b). A similar trend of hydraulic performance was observed in previous studies [35,36]. The plateau permeability suggests that the foulant accumulation rate equalled the foulant dispersal rate due to the fouling control mechanism [37,38]. Extending the filtration time was expected to decline the permeability at a much slower rate.

Figure 5b shows that introducing 30 rpm of rotation to the static disk disk enhanced the membrane permeability from 96.4 ± 3.6 to 132.7 ± 2.1 Lm^−2^h^−1^bar^−1^, corresponding to an increment of about 27%. The finding demonstrated that the disk rotation substantially increased the permeability when comparing the first two data of 0 and 30 rpm. The *p*-value (of 0.0047) corresponding to the F-statistic (11.9600) was obtained from a one-way ANOVA analysis to describe the statistical dependence of permeability on the rotational velocity of 30 rpm. This *p*-value was lower than 0.05, suggesting that a disk rotation of 30 rpm increased permeability significantly. However, it is unclear at what rotating speed the rotary spacer started to offer significant membrane fouling control since the motor employed in our set-up was limited to 30 rpm as the lowest value. This finding suggests the efficiency of the rotating spacer as a fouling control mechanism for the filtration system, as reported earlier for different applications and system designs [39].

Figure 6 shows that the rotation of the disk spacer generated shear to the feed liquid, which induced a secondary flow of liquid, creating shear stress near the static membrane surface. This condition helped reduce the fouling propensity and provided the force to scour off the deposited foulant and prevent the foulant accumulation.

Observation of the effect of rotation speed in Figure 5 shows that a higher rotating speed led to a higher permeability, which agreed with the shear rate provided on the membrane surface profiled in Figure 6B,C. The membrane reached the permeability of 144.6 ± 3.6 Lm^−2^h^−1^bar^−1^ at the rotation speed of 40 rpm and peaked at 175.0 ± 5.0 Lm^−2^h^−1^bar^−1^ at 50 rpm (the highest speed evaluated in this study). The overall 38% permeability increment was similar to that achieved by Sarkar et al. [25], who studied the filtration of casein whey using a rotating disc membrane. Although it was expected that the flux permeation would be enhanced with a further increment of rotating speed due to a higher shear rate, further increase of rotational speed beyond 40 rpm (≤50 rpm) showed minimal impact on the permeability. The efficacy of spacer rotation in enhancing permeability was identical to the rotating module reported elsewhere [28], in which the rotating motion helped reduce the foulant-cake-layer thickness.

Multiple means comparison using post hoc Tukey HSD test found a significant permeability increment only when the difference between the disk rotation speed was at least 10 rpm under 30–45 rpm. Significant permeability increments were obtained when the rotation speed was increased from 30–40 rpm, 30–45 rpm, 30–50 rpm, 35–45 rpm, or 35–50 rpm, all with Tukey HSD *p*-values of less than 0.015. It indicated a significant increment at the rotational speed step of >10 rpm. At high rotation speed values above 40 rpm, the effect of disk rotation speed in increasing permeability was low, which is consistent with the wall share profile shown in Figure 6A. It might be due to the small rotating speed range compared to that used in other reported studies [39,40,41,42] and the sidewall effect that led to a substantial increment of shear near the edge of the membrane. This finding implies that a very high spacer rotation speed is not required, and the system can work effectively under low rotation speeds. This condition benefits energy input since a higher rotation speed results in higher operational costs due to higher energy consumption [40].

### 3.3. Effect of Membrane-to-Disk Gap

Figure 7 depicts the effect of the membrane-to-disk gap on the permeability of the oil/water emulsion filtration evaluated under a fixed rotating speed of 50 rpm. It shows that a higher membrane-to-disk gap reduced the permeability. When the membrane-to-disk gap was increased from 0.5 to 1.0, 1.5, and 2.0 cm, the membrane showed decreasing steady-state permeabilities, i.e., from 156.0 ± 5.7 to 146.4 ± 3.6, 141.7 ± 1.0, and 91.1 ± 1.8 Lm^−2^h^−1^bar^−1^, respectively. Smaller membrane-to-disk gaps led to higher shear stress experienced by the membrane surface because the shear stress is inversely proportional to the gap. The high velocity of the liquid nearby the membrane surface also exerted the drag force that lifted the foulant from the membrane surface. This way, the buildup of an oil layer as a cake layer as foulant could be limited [43].

Multiple means comparison using post hoc Tukey HSD test found an insignificant difference in permeabilities between gaps of 0.5 vs. 1.0 cm (Tukey HSD *p*-value of 0.740). The differences in permeabilities became significant, as shown by the Tukey HSD *p*-value of less than 0.0059. A small disk-to-membrane gap below 1.0 cm would result in a statistically similar permeability. Hence, to allow optimum membrane fouling control using the developed rotary spacer system, it is desirable to place the membrane closer than 1.0 cm from the disc. In addition, it is worth mentioning that a small gap between membrane and disk is also favourable in a full-scale unit from packing density [11]. A follow-up study will describe the membrane surface’s fluid dynamic mechanism. The previous reports show that a CFD simulation would visualize the shear stress profile on the membrane surface [44,45].

### 3.4. Specific Energy Consumption for Fouling Control

Apart from demonstrating higher filtration throughput, a system for membrane fouling control is expected to work under a low energy input. Figure 8 shows the specific energy consumption associated with the membrane fouling system of a projected full-scale membrane using a rotary spacer filtration system. The energy was utilized for mechanically rotating the spacers. The estimation was done using the filtration data presented in Figure 5. The specific energy input for the operating mechanical rotation of the spacers was in the range of 1.1 to 4.1 × 10^−3^ kWhm^−3^ permeate. The estimations were done for the average linear velocities of 0.3 to 0.4 ms^−1^, corresponding to radial disk velocities of 30 to 50 rpm applied in the experiments or the hypothetical spacer’s radial disk velocities of 3.1 to 5.2 rads^−1^.

The specific mechanical energy inputs were for a system with higher velocity. However, it does not mean that the system without or with low rotation speeds would offer the lowest overall energy consumption. Figure 8 implies that higher disk rotation would lead to higher permeability and thus lower membrane area and membrane investment costs. The permeability is also closely associated with system footprint, cleaning frequencies, etc. The conditions required to achieve the lowest overall capital and operational expenditures need to be optimized further.

Compared with data in the literature, the energy consumption for membrane fouling control using the rotary spacer system was lower by two orders of magnitude. Full-scale submerged membrane bioreactors typically consume energy for membrane cleaning using air bubbling around 40–60% of the overall energy inputs (0.4–0.6 kWhm^−3^), corresponding to 0.16 to 0.36 kWhm^−3^. To demonstrate the degree of inefficiency of the current aeration system, optimization of the submerged systems through design operation and equipment could reduce the energy by up to 81% [46]. The energy consumption associated with membrane fouling control for air-optimized bubbling and panel switching was 0.087−0.103 kWh m^−3^ [7]. For a more fouling-prone system of anaerobic membrane bioreactor, rigorous analysis of the energy consumption showed that the energy demand associated with gas sparging was at a range of 1.3–1.4 kWhm^−3^ and 0.2 ± 0.1 kWhm^−3^ for the flat sheet and the hollow fibre membranes, respectively [47]. The high energy demand for the flat-sheet membrane was attributed to the high gas sparging intensity coupled with a low flux (≈7 Lm^−2^h^−1^). Using a flat-sheet membrane, the optimized system consumed 0.7–5.7 kWh/m^3^ for the gas sparging and recirculation [48]. In another review report, the energy consumptions of pilot-scale anaerobic membrane bioreactors associated with the control of membrane fouling via gas recirculation were reported to be 0.04 to 1.35 kWhm^−3^ with an average of 0.41 kWhm^−3^ [49].

The finding on energy assessment suggests that the energy input for spacer rotation is highly promising. The spacer rotation can also solve the issue of clogging in the feed flow channel when treating feed with high solid by sweeping it away when it adheres in the dead-zone near the membrane surface, as applied in a rotating biological membrane contactor [50]. However, incorporating spacers would lower module packing density and enlarge the plant footprint. The key to overcoming this issue is developing highly thin spacers and placing them very close to the membrane surfaces, as proposed earlier in the vibrating spacer system [14]. Moreover, this system can be further explored for different applications, particularly on challenging and viscous feeds such as anaerobic sludge or hydrolysate filtration [44]. Applying traditional air bubbling would be less effective. The system’s performance is expected to be further enhanced when equipped with less fouling-prone membranes, such as the one developed recently through the co-induced phase separation method [51] or sulfonated carbon soot-polysulphone [52].

## 4. Conclusions

This study demonstrated a rotary spacer system as an effective and energy-efficient method for controlling membrane fouling in oil/water emulsion filtration. The rotation of the spacer placed between two adjacent membrane panels induces a secondary flow of the feed liquid that imposes shear rate and lift force to scour off and remove the foulant from the membrane surface, as also proven by the CFD simulation. Rotating the spacer at up to 50 rpm (v = 0.44 ms^−1^) increased the oil/water emulsion permeability from 126.8 ± 2.1 for the static system to 175.5 ± 2.7 Lm^−2^h^−1^bar^−1^ (rotated at 50 rpm). Since the shear rate is inversely proportional to the gap, increasing spacer-to-membrane gaps from 0.5 to 2.0 cm at a constant rotational speed lowered the permeability from 175.5 ± 2.7 to 126.7 ± 2.0 Lm^−2^h^−1^bar^−1^. Remarkably, low energy consumptions were required to drive the mechanical rotation of the spacer. It is in the range of 1.1 to 4.1 × 10^−3^ kWhm^−3^ permeate for systems run at average linear velocities of 0.3 to 0.4 ms^−1^. The energy inputs were two orders of magnitude lower than that of established aeration systems, suggesting its competitiveness for other applications. The design of the half-circular disk in the set-up my affect the result significantly and needs to be addressed in the future. In addition, a long-term test of the system’s efficacy need to be done to beter gauge its full potential. Finally, a system with the smallest disk-to-membrane gap could offer the best solution of achieving a high throughput with lower footprint.

## Figures and Tables

**Figure 1 membranes-12-00554-f001:**
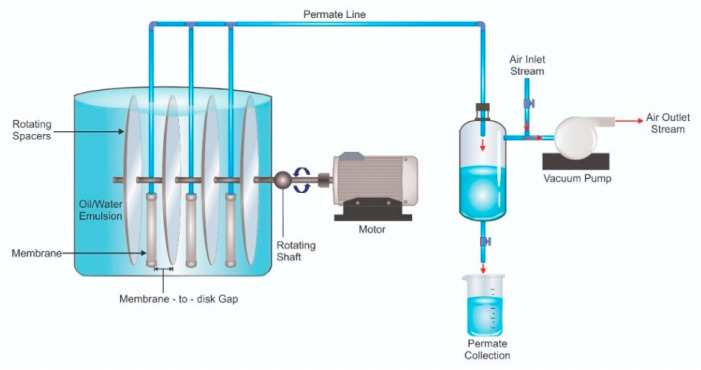
Illustration of laboratory-scaled rotating spacer submerged membrane filtration system. In the actual set-up, the single panel had a half-round shape and membrane sheets with total surface area of 0.2 m^2^ were glued on both sides of the panel.

**Figure 2 membranes-12-00554-f002:**
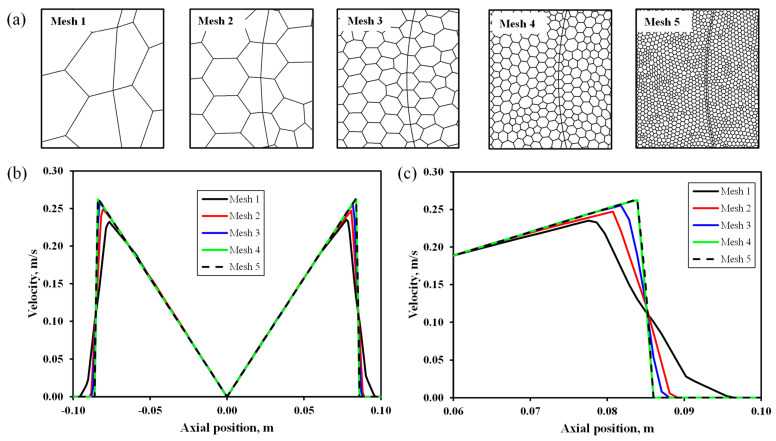
Illustration of meshing quality examples for grid sensitivity analysis showing (**a**) the grids and (both (**b**,**c**)) the sensitivity.

**Figure 3 membranes-12-00554-f003:**
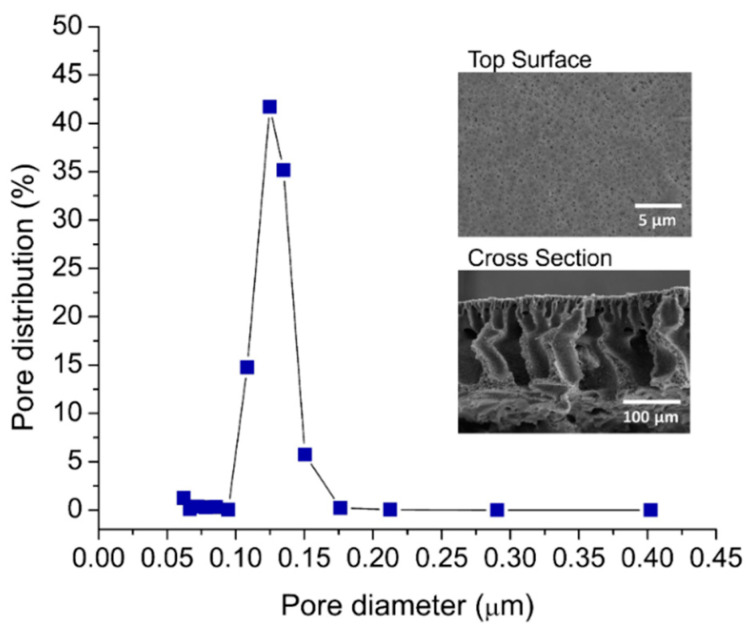
The pore diameter distribution of the developed PVDF membrane. The insets show their morphological images.

**Figure 4 membranes-12-00554-f004:**
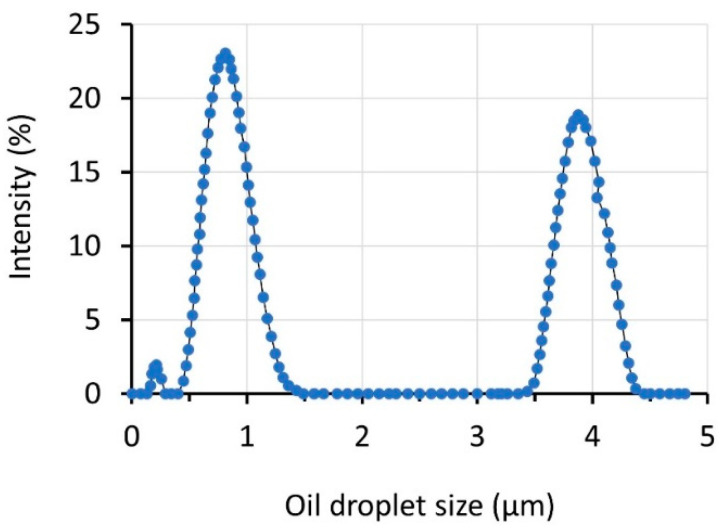
Size distribution of oil droplet in oil/water emulsion sample.

**Figure 5 membranes-12-00554-f005:**
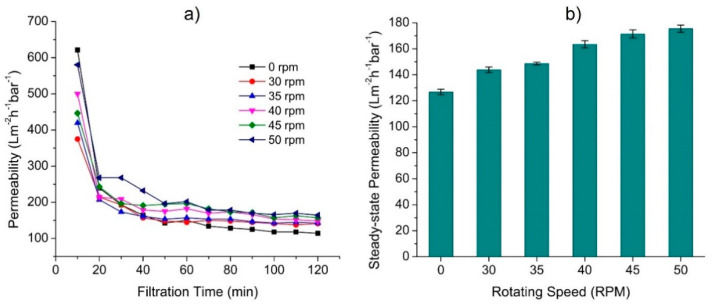
The effect of rotating speed on the membrane permeability permeance showing (**a**) the evolution of permeability as function of filtration time, and (**b**) the steady-sate permability value taken as the average of the final 30 min filtration. The system was run at a membrane-to-disk gap of 1.5 cm.

**Figure 6 membranes-12-00554-f006:**
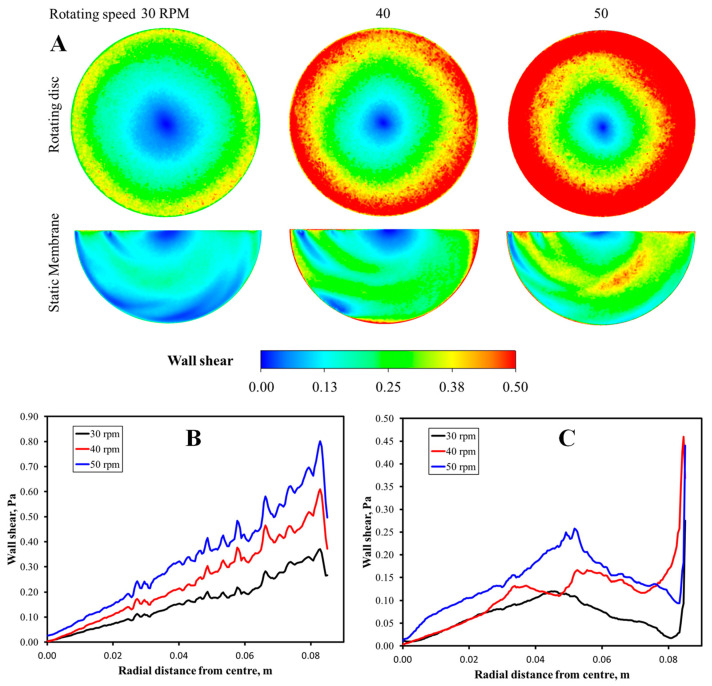
(**A**) Visualization of wall shear and its profile as a function of the radial distance on the surface of (**B**) rotating spacer and (**C**) the static membrane surface.

**Figure 7 membranes-12-00554-f007:**
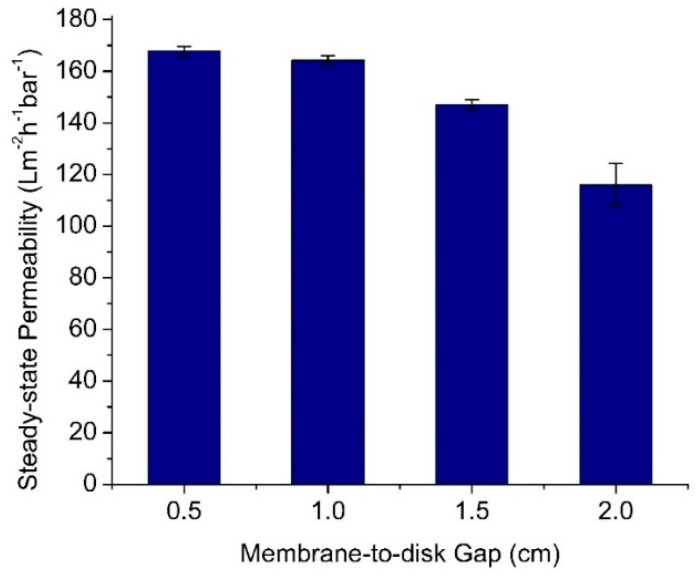
The effect of the membrane-to-disk gap on the permeability. A narrow gap offered a higher hydraulic throughput.

**Figure 8 membranes-12-00554-f008:**
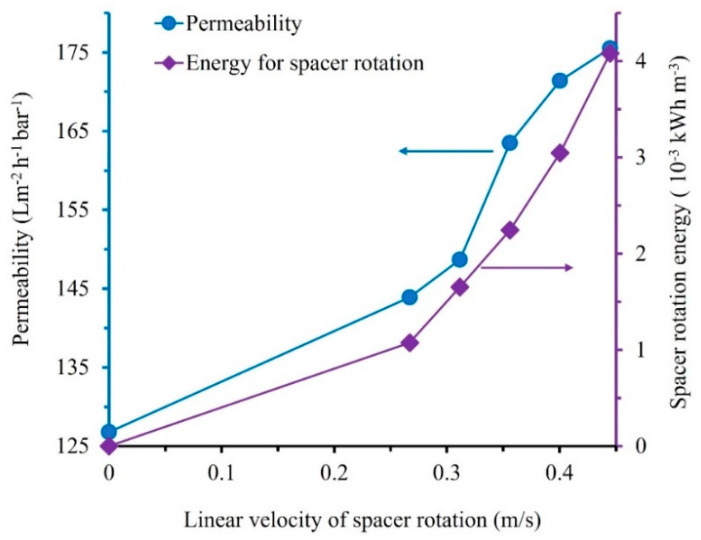
The energy consumption of the rotary spacer system operated under variable rotation speeds.

**Table 1 membranes-12-00554-t001:** Summary of meshing qualities for the sensitivity analysis.

Mesh	Size (mm)	No. of Cells	
		Tetrahedral	Polyhedral
1	20.0	403,411	74,510
2	10.0	408,215	76,023
3	5.0	456,440	86,953
4	2.5	780,897	152,042
5	1.0	4,127,764	779,710

## Data Availability

Not applicable.

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
