# Peer review of "A Rotary Spacer System for Energy-Efficient Membrane Fouling Control in Oil/Water Emulsion Filtration"

_membranes, 2022, doi:10.3390/membranes12060554_

Round 1

Reviewer 1 Report

The authors presented an innovative method of preventing the unfavorable phenomena of membrane fouling, namely the rotating spacer system. The work is of a research nature with development potential. I appreciate that the authors presented in the work a further research path to be performed in order to obtain higher efficiency and repeatability in real conditions. I am in favor of publishing the work in Membranes after minor correction.

  1. Section 2.3. „The membrane panel had an effective surface area of 0.2 m2 in the shape of a segment of a half-circle with a diameter of 17 cm, which was connected to a permeate collecting system.” - does the membrane panel mean one sheet of membrane (dimensions?) with an active area of 0.2 m2? 3 such panels were used?
  2. Section 2.3. “The permeate was 153 collected during relaxation time of 10 mins of the filtration cycle, composing nine mins 154 and one min of relaxation.” - this sentence is unclear, because it gives the time of 9 minutes and 1 minute, of which 9 minutes what is it? additionally in the next chapter in the sentence: "The permeate volume was measured for each filtration cycle during the relaxation period." - the permeate was collected in what time, 10 minutes or 1 minute?
  3. On the Fig 2a – the mesh size 4 is larger than the mesh size 3?

Reviewer 2 Report

The paper describes the development of a rotary spacer system for an submerged membrane filtatration set-up. The rotation of the spacers, improved the permeability of the membrane due to an increase in shear forces on the membrane surface. The research is of average novelty, quality, and scientific sound. I propose to accept this paper with minor revisions.

My comments can be found below. Please be aware that they seem many but most of them can be fixed rather fast.

In general, grammar and spelling could be improved.

Introduction:
Line 47,48: I find it inappropiate to make 5 self-citations (references 5-9) for such general statements, and recommend to adjust this.
Line 59: Module name is missing

Materials and Methods:
Generally: The analytical methods should be described in more detail in order to be able to understand and reproduce the measurements.
Line 124: Company name is wrong. I guess you meant Freudenberg Filtrations Technologies?
Line 143: Please provide more information about the feed. I am missing information about the volume of feed solution, and the volume of the samples analyzed. Furthermore, how did you analyze the droplet size distribution, what method did you use, what assumptions did you make?
Line 153: The SI unite is "min" not "mins."
Line 154: I don't understand how/when you collected the permeate.
Figure 1: The depiction of the membrane (hollow fiber module?) does not fit to the description of the membrane (half-circle). Please adjust
Line 189: What is the Tukey HSD post hoc analysis, please add a reference or explain in more detail.
Subchapter 2.7: This subchapter and Figure 2 could be shortened or partly put in the supplementary materials as it is a mix between methods, results and discussion but only indirectly related to the research work.

Results and Discussion
General comment: The two decimal digits are pretend a high accuracy of the work, but with slight changes to the feed solutions this could be changed. Hence, I recommend to remove the decimal digits or state only 1. Furthermore, how did you determine the deviations? Did you perform each experiment several time? This is unclear to me but must be stated.
Line 312: I think you mixed up rpm with cm.
Line 366: Why do you highlight the difference between air bubbling and air sparging? What is the difference?
Line 373: How does that work? Also, please state the original reference and not the review.

Conclusions:
Line 391: Increasing the rotation speed from what to up to 50 rpm?

Reviewer 3 Report

Review of the article entitled: A rotary spacer system for energy-efficient membrane fouling control in oil/water emulsion filtration

Authors: Normi Izati Mat Nawi, Afiq Mohd Lazis, Aulia Rahma, Muthia Elma, Muhammad Roil Bilad, Nik Abdul Hadi Md Nordin, Mohd Dzul Hakim Wirzal, Norazanita Shamsuddin, Hazwani Suhaimi, Norhaniza Yusof

In my opinion, the reviewed manuscript fits the scope of Membranes journal. The paper is well organized and written. The Authors presented the results in the concise form, and the hypotheses were confirmed by proper explanations and wide discussion with the literature. I recommend the manuscript to be published in Membranes journal in the present form.

Author Response

Thank you for the positive evaluation of our work. Some minor revisions have been performed in the revised manuscript to accommodate comments from other reviewers.